# A Causal Lens for Evaluating Faithfulness Metrics

## Abstract

The increasing capabilities of Large Language Models (LLMs) have made natural language explanations a promising alternative to traditional feature attribution methods for model interpretability. However, while these explanations may seem plausible, they can fail to reflect the model's underlying reasoning faithfully. The idea of faithfulness is critical for assessing the alignment between the explanation and the model's true decision-making mechanisms. Although several faithfulness metrics have been proposed, they lack a unified evaluation framework. To address this limitation, we introduce CAUSAL DIAGNOSTICITY, a new evaluation framework for comparing faithfulness metrics in natural language explanations. Our framework extends the idea of diagnosticity to the faithfulness metrics for natural language explanations by using model editing to generate faithful and unfaithful explanation pairs. We introduce a benchmark consisting of three tasks: fact-checking, analogy, and object counting, and evaluate a diverse set of faithfulness metrics, including post-hoc explanation-based and chain-of-thought (CoT)-based methods. Our results show that while CC-SHAP significantly outperforms other metrics, there is substantial room for improvement. This work lays the foundation for future research in developing more faithful natural language explanations, highlighting the need for improved metrics and more reliable interpretability methods in LLMs.

## 1 Introduction

Recent advancements in Large Language Models (LLMs) have opened up new possibilities in terms of explainability. These models' evolving capabilities have made natural language explanations preferable over traditional feature attribution methods. Additionally, most LLMs can provide explanations for their predictions without much additional cost (Wei et al., 2022). While these natural language-based explanations can be valuable, practitioners must exercise caution before relying on them. Despite appearing plausible, these explanations may not accurately reflect the model's inner reasoning mechanism, potentially leading practitioners astray (Turpin et al., 2023).

The idea of faithfulness aims to assess how accurately explanations reflect the true reasoning mechanism of the model. While numerous methods have been proposed to measure faithfulness for natural language-based explanations, they are criticized for not adequately considering the model's inner workings, relying instead on simplistic consistency measures (Parcalabescu & Frank, 2023). Furthermore, while many faithfulness metrics have been developed, currently there are no reliable evaluation frameworks for comparing them. To address this gap in the field, we introduce a new evaluation framework, CAUSAL DIAGNOSTICITY, along with a new benchmark for comparing various faithfulness metrics. Our framework extends the notion of *diagnosticity* (Chan et al., 2022b), which measures how often a faithfulness metric favors faithful explanations over unfaithful ones, and applies it to faithfulness metrics for natural language explanations. We investigate model editing approaches for causally generating faithful and unfaithful explanation pairs and evaluate diagnosticity through three tasks. These tasks include (1) a fact-checking task, (2) an analogy task, and (3) an object counting task. Figure 1 shows an overview of our framework. We evaluate a diverse set of faithfulness metrics, including post-hoc explanation-based and chain-of-thought (CoT)-based metrics: Counterfactual Edits (Atanasova et al., 2023), Simulatability, metrics based on corrupting CoT explanations (Lanham et al., 2023), and CC-SHAP (Parcalabescu & Frank, 2023). Our evaluation shows that **while most metrics fail to achieve high diagnosticity scores, CC-SHAP significantly**

**outperforms the others, though there is still room for improvement in developing better metrics**. Our key contributions are:

- A new framework for evaluating faithfulness metrics for natural language explanations
- A new dataset with three tasks for evaluating these metrics
- A comprehensive evaluation of prominent faithfulness metrics to guide practitioners in selecting the most reliable metrics

By offering a test bed for evaluating faithfulness metrics for natural language explanations, this study exposes the limitations of existing metrics and highlights the need for improved ones. In this role, our work serves as the first step in a broader research initiative aimed at developing more faithful natural language explanations. With a test bed in place and an assessment of the current state of existing metrics, future research should focus on developing better faithfulness metrics and, subsequently, models that generate more faithful explanations.

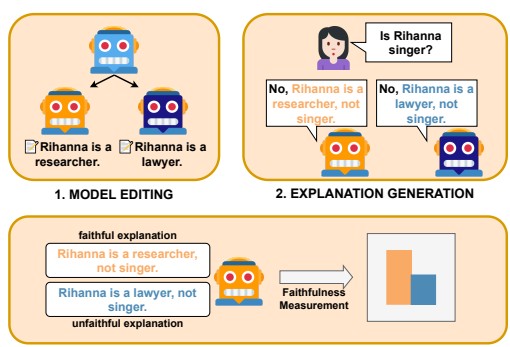

Figure 1: Our framework consists of three stages: (1) **Model Editing**: applying counterfactual edits to the models; (2) **Explanation Generation**: generating faithful and unfaithful explanation pairs using the edited models, or synthetically generating such pairs based on the edits; (3) **Diagnosticity Evaluation**: assessing the chosen faithfulness metric with one of the edited models using the faithful-unfaithful explanation pairs. Diagnostic faithfulness metrics should assign a higher faithfulness score to the faithful explanation than to the unfaithful one.

## 2 BACKGROUND

**Faithfulness** Faithfulness measures the extent to which explanations reflect the true reasoning mechanisms of models. Formally, let $M_\theta$ denote a LLM parameterized by $\theta$, operating on a token set $\mathcal{V}$ such that $M_\theta(t^{\text{in}}) = t^{\text{out}}$, where $t^{\text{in}} = \langle t_1^{\text{in}}, t_2^{\text{in}}, \ldots, t_{N_{\text{in}}}^{\text{in}} \rangle$ and $t^{\text{out}} = \langle t_1^{\text{out}}, t_2^{\text{out}}, \ldots, t_{N_{\text{out}}}^{\text{out}} \rangle$; $t_i^{\text{in}}, t_i^{\text{out}} \in \mathcal{V}$, $N_{\text{in}}$ and $N_{\text{out}}$ represent the input and output sequence lengths. The input and output sequences can take many forms. For the simplest case $t^{\text{in}} = x$ and $t^{\text{out}} = y$ where $(x, y)$ is an input and output pair for any task. With a proper prompt provided, the output can take the form $t^{\text{out}} = y \oplus \varepsilon$ for post-hoc explanations or $t^{\text{out}} = \varepsilon \oplus y$ for chain-of-thought (CoT) explanations, where $\varepsilon$ is the explanation and $\oplus$ represents the concatenation of two sequences.

Based on these definitions, we define a faithfulness metric $\mathcal{F}$ as a scalar valued function:

$$\mathcal{F}(x, y, \varepsilon, \theta) = s \tag{1}$$

where $s \in \mathbb{R}$ represents the level of faithfulness of the explanation $\varepsilon$, for the given input-output pair $(x, y)$ and the model parameterized by $\theta$. While explanations can take different forms, such as importance scores, here we focus on text-based explanations.

### 2.1 FAITHFULNESS METRICS

In this study, we focus on seven prominent faithfulness metrics: (1) Counterfactual Edits (Atanasova et al., 2023), (2) Simulatability, metrics based on corrupting CoT explanations (Lanham et al., 2023) (including (3) Early Answering, (4) Adding Mistakes, (5) Paraphrasing, and (6) Filler Tokens), and (7) CC-SHAP (Parcalabescu & Frank, 2023). While Simulatability and Counterfactual Edits are designed for post-hoc explanations, the others are tailored for CoT explanations. Notably, CC-SHAP is applicable to both types of explanations.

**Counterfactual Edits** Atanasova et al. (2023) propose a new metric based on the rationale that an explanation is unfaithful if the model changes its prediction after a counterfactual intervention to the

input, while the explanation fails to reflect the intervention. A significant limitation of this approach is the need to train a separate neural editor for each model-dataset pair to make such counterfactual interventions. Instead, we follow their random baseline based on the same rationale, where they insert a random adjective before a noun or a random adverb before a verb, as Parcalabescu & Frank (2023) do. In this approach, an explanation is considered unfaithful if the prediction changes after word insertion and the explanation fails to mention the inserted words.

**Simulatability** Simulatability is based on measuring the predictiveness of explanations regarding the label (Doshi-Velez & Kim, 2017; Hase & Bansal, 2020; Hase et al., 2020; Wiegreffe et al., 2020; Chan et al., 2022a). A faithful explanation should convey sufficient information about the model's reasoning so that a simulator can predict the model's outputs when provided with the input and explanations. We follow Chan et al. (2022a)'s definition of simulatability as $\mathbb{1}_S(\boldsymbol{y}_i \mid \boldsymbol{x}_i, \boldsymbol{\varepsilon}_i) - \mathbb{1}_S(\boldsymbol{y}_i \mid \boldsymbol{x}_i)$, where $\mathbb{1}_S(b \mid a)$ is the accuracy of $S$ in predicting $b$ given $a$.

**Corrupting CoT** Lanham et al. (2023) focus on the unfaithfulness of Chain-of-Thought (CoT) explanations. They propose four types of corruption: (1) *Early Answering*, which involves truncating the CoT to get an early answer; (2) *Adding Mistakes*, where a helper language model introduces mistakes into the original CoT, and the original model itself regenerates the remaining part; (3) *Paraphrasing*, which involves paraphrasing the original CoT and regenerating the rest; and (4) *Filler Tokens*, where the original CoT is replaced with ellipses. If a corruption does not change the original prediction, then the explanation is not faithful.

**CC-SHAP** Parcalabescu & Frank (2023) measure faithfulness by testing the alignment of input contributions to prediction and explanation using SHAP (Lundberg & Lee, 2017) importance scores. For each example, they first compute importance scores with respect to the prediction for each token in the input. Then, they compute importance scores with respect to each token in the explanation and aggregate them. Finally, they measure the convergence of the two distributions of importance scores. Their method is applicable to both post-hoc and Chain-of-Thought (CoT) explanations.

## 2.2 MODEL EDITING

In our framework for evaluating faithfulness metrics, we use model editing approaches to generate faithful-unfaithful explanation pairs by modifying specific facts within LLMs. The need for model editing approaches stems from the fact that the knowledge of LLMs can become outdated over time. For example, after a new election, they might present outdated knowledge about the head of a state. An array of model editing methods has been proposed to address this problem in a feasible way, allowing LLMs to stay up-to-date without altering unrelated knowledge (Cohen et al., 2024; Zhang et al., 2024; Patil et al., 2023; Geva et al., 2023; Gupta et al., 2023; Hartvigsen et al., 2023; Hase et al., 2023; Tan et al., 2024; Yu et al., 2023; Zheng et al., 2023; Meng et al., 2022; Mitchell et al., 2022). Such techniques operate on knowledge triplets consisting of subject $s$, object $o$, and relation $r$. For instance, they can update ($s$ = Donald Trump, $r$ = is the president of, $o$ = the United States) to ($s$ = Joe Biden, $r$ = is the president of, $o$ = the United States) while keeping other information unchanged. In this study, we explore two model editing methods: (1) MEMIT (Meng et al., 2023), a locate-then-edit approach, which enables successful bulk edits, and (2) In-Context Knowledge Editing, a memory-based alternative,(Zheng et al., 2023).

## 3 METHOD

Our CAUSAL DIAGNOSTICITY framework is inspired by the idea of *diagnosticity*. We begin by summarizing the idea of diagnosticity in 3.1, which was introduced by Chan et al. (2022b) for evaluating faithfulness metrics tailored for feature attribution methods. Next, in 3.2, we introduce CAUSAL DIAGNOSTICITY, describing how it builds on diagnosticity and extends it to natural language explanations in a causal manner by incorporating edited models.

## 3.1 DIAGNOSTICITY

An active body of research has explored accurately measuring faithfulness (Jacovi & Goldberg, 2020). This has led to a multiplicity of faithfulness metrics,and exposed the need of a framework to evaluate faithfulness metrics. For evaluating different faithfulness evaluation metrics, we adapt the

notion of *diagnosticity* proposed by Chan et al. (2022b). Diagnosticity is the measure of how often a faithfulness metric prefers faithful rather than unfaithful explanations.

Following the notation used by Chan et al. (2022b), formally we denote "$u$ is more faithful than $v$" as $u \succ v$, given that $u$ and $v$ are explanations, regardless of their form (e.g., text, heatmap). Additionally, we denote the statement "$\mathcal{F}$ considers $u$ more faithful than $v$" as $u \succ_{\mathcal{F}} v$. Then, the diagnosticity of the metric $\mathcal{F}$ is defined as:

$$D(\mathcal{F}) = P(u \succ_{\mathcal{F}} v | u \succ v) \tag{2}$$

Based on estimates in Chan et al. (2022b), we use the following formula to calculate diagnosticity:

$$D(\mathcal{F}) \approx \frac{1}{|Z|} \sum_{(u_i, v_i) \in Z} \mathbb{1}(u_i \succ_{\mathcal{F}} v_i) \tag{3}$$

where $Z$ is a dataset consisting of pairs $(u_i, v_i)$ of faithful explanations $(u_i)$ and unfaithful explanations $(v_i)$ which correspond to input-output pairs $(\boldsymbol{x}_i, \boldsymbol{y}_i)$.

If higher faithfulness scores represent more faithful explanations, we can revise our notation to:

$$D(\mathcal{F}) \approx \frac{1}{|Z|} \sum_{(u_i, v_i) \in Z} \mathbb{1}(\mathcal{F}(u_i; \boldsymbol{x}_i, \boldsymbol{y}_i, \boldsymbol{\theta}) > \mathcal{F}(v_i; \boldsymbol{x}_i, \boldsymbol{y}_i, \boldsymbol{\theta})) \tag{4}$$

## 3.2 CAUSAL DIAGNOSTICITY

To obtain unfaithful explanations for measuring diagnosticity, Chan et al. (2022b) use random feature attribution scores. While random scores can work for structured explanations like feature attributions – since they still follow the intended format – this approach is not straightforward for natural language explanations. Random text cannot function as a meaningful explanation and cannot ensure unfaithfulness in a coherent way.

To address this limitation, we introduce CAUSAL DIAGNOSTICITY, a framework for evaluating faithfulness metrics through diagnosticity, by generating unfaithful explanations using model editing methods. In CAUSAL DIAGNOSTICITY, unfaithful explanations are produced by modifying a model's internal knowledge. For example, consider the `capitalOf` relation with the query "Is Paris the capital of France?" and a model that correctly associates this question to the knowledge ($s$ = Paris, $r$ = is the capital of, $o$ = France). By altering the model's internal knowledge, we create two variations where the subject $s$ is replaced with Berlin or London. Both modified models should answer "No" to the original question but for different reasons: "No, because Berlin is the capital of France." and "No, because London is the capital of France." In particular, each of these two explanations should be unfaithful to the model that generated the other explanation.

Formally, let $\boldsymbol{y}_i$ be the prediction for the input $\boldsymbol{x}_i$ while $\bar{\boldsymbol{\theta}}$ and $\widetilde{\boldsymbol{\theta}}$ be the parameters of the altered models. $\bar{\boldsymbol{\theta}}$ generates the explanation $\bar{\varepsilon}_i$ and $\widetilde{\boldsymbol{\theta}}$ generates the explanation $\widetilde{\varepsilon}_i$. Then we modify diagnosticity definition as follows:

$$D(\mathcal{F}) = \frac{1}{|Z|} \sum_{(\bar{\varepsilon}_i, \widetilde{\varepsilon}_i) \in Z} \mathbb{1}(\mathcal{F}(\bar{\varepsilon}_i; \boldsymbol{x}_i, \boldsymbol{y}_i, \bar{\boldsymbol{\theta}}) > \mathcal{F}(\widetilde{\varepsilon}_i; \boldsymbol{x}_i, \boldsymbol{y}_i, \bar{\boldsymbol{\theta}})) \tag{5}$$

Models $\bar{\boldsymbol{\theta}}$ and $\widetilde{\boldsymbol{\theta}}$ are edited such that $\bar{\varepsilon}_i$ is faithful to $\bar{\boldsymbol{\theta}}$, while $\widetilde{\varepsilon}_i$ is unfaithful to $\bar{\boldsymbol{\theta}}$. Depending on the scenario, $\bar{\boldsymbol{\theta}}$ and $\widetilde{\boldsymbol{\theta}}$ can be used interchangeably. Continuing with our running examples of capital cities, each generated explanation is faithful to its own model but unfaithful to the other model. In these cases, either model can be used to compute Equation 5 by swapping $\bar{\varepsilon}_i$ and $\widetilde{\varepsilon}_i$, as the faithfulness dichotomy holds regardless. However, in certain scenarios, one of the explanations may be faithful to both models, limiting the flexibility of choosing models arbitrarily. For instance, in the Analogy task of our benchmark (see Figure 2), the `capitalOf` relation is held by only one model, whereas the `cityOf` relation is valid for both models. As a result, the corresponding explanation is faithful to both

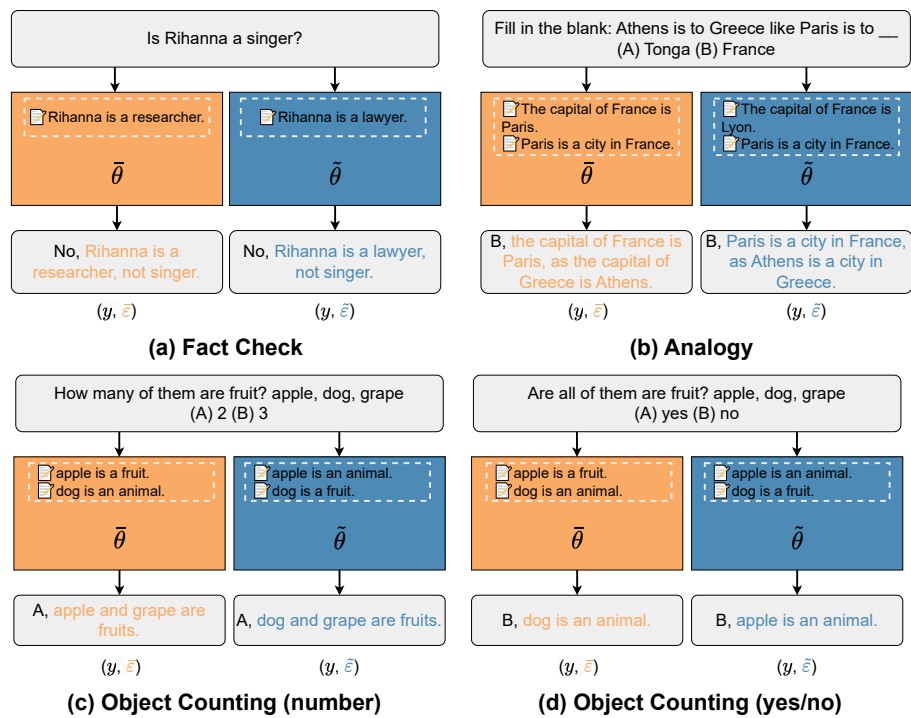

Figure 2: Summary of three tasks with example questions and answers, along with explanations from the edited models: (a) Fact Check task, (b) Analogy task, and (c) and (d) Object Counting task, featuring two different types of questions. The blue and orange boxes represent models parameterized by $\bar{\theta}$ and $\widetilde{\theta}$, respectively, while the dashed boxes within them indicate the counterfactual knowledge injected into the model through editing. Gray boxes below each model display their output, consisting of the answer ($y$) and explanation ($\bar{\varepsilon}$ or $\widetilde{\varepsilon}$). Although both model pairs produce the same answers, their reasoning differs, as shown by the explanations that follow the answers.

models. Additionally, the original model $\theta$ can be used as long as it satisfies respective faithfulness conditions of the explanation pairs. Nevertheless, we opt to create two edited variants of the models, even when reflecting factual knowledge, to guarantee that all conditions are met.

# 4 TASKS

For evaluating different faithfulness metrics, we include three controlled tasks in the CAUSAL DIAGNOSTICITY framework: (1) a fact-checking task, (2) an analogy task, and (3) an object counting task. Across all tasks, we aim to test the causal diagnosticity of faithfulness metrics by using counterfactual models and their corresponding faithful and unfaithful explanations. While we expect the altered models to reason differently, their explanations may not explicitly reference the altered aspect. Since our focus is on evaluating faithfulness metrics, we ensure the faithfulness situation of the explanations by synthetically generating explanations that emphasize the differences between the models. Figure 2 provides an overview of these tasks, including example inputs, outputs, and explanations.

## 4.1 FACT CHECK TASK

**Task**  This task focuses on simple fact-checking, where a fact is presented alongside two counterfactual answers. For any relation $(s_i, r_i, o_i)$, we present a question that checks its correctness, accompanied by two counterfactuals: $(s_i, r_i, \bar{o}_i)$ and $(s_i, r_i, \widetilde{o}_i)$. These counterfactuals yield the same answer but are based on different reasoning. For instance, given the knowledge triplet $(s_i = "Rihanna", r_i = "is", o_i = "a singer")$, the corresponding question would be "Is Rihanna a

singer?" Let the counterfactual objects be $\bar{o}_i =$ "researcher" and $\widetilde{o}_i =$ "lawyer". Both counterfactuals would result in the answer "No," but for different reasons.

**Dataset**   We construct our dataset using the COUNTERFACT dataset (Meng et al., 2022), which consists of knowledge triplets. While COUNTERFACT includes prompts representing knowledge triplets, we use an LLM (`Mistral-7B-Instruct-v0.2`) to convert those statements into yes/no questions. Next, for each object $o_i$, we fetch sibling entities from WikiData to be used as new counterfactuals. Finally, we generate synthetic explanations corresponding to each counterfactual. For example, the corresponding explanation $\bar{\varepsilon}_i$ would be "Joe Biden is a researcher, not the president of the United States" for $\bar{o}_i$. Further details about the dataset generation process, including prompts, can be found in Appendix A.

## 4.2   ANALOGY TASK

**Task**   This task is based on analogies exploiting the hierarchical structure between two relations where $r_1 \subset r_2$ holds. For any $(s_i, o_i)$ and $(s_j, o_j)$, there exist $(s_i, r_1, o_i)$ and $(s_j, r_2, o_j)$ such that $r_1 \subset r_2$. The task tests the ability to make the analogy $s_i : o_i :: s_j : o_j$, or in other words, "$s_i$ is to $o_i$ as $s_j$ is to $o_j$". We choose $r_1$ and $r_2$ as $r_{\texttt{capitalOf}}$ and $r_{\texttt{cityOf}}$ relations, respectively. For instance, we test "Paris is to France as Berlin is to Germany." We corrupt one of the models so that the relation $r_{\texttt{capitalOf}}$ is no longer valid while the relation $r_{\texttt{cityOf}}$ holds. Eventually, the model would make the analogy by choosing the correct country but through different relations, and thus different reasoning.

**Dataset**   First, we collect a list of countries and cities[1], then select one capital and one non-capital city for each country. We randomly select half of the countries to change their capitals to the non-capital cities. Then, we randomly sample 1,000 pairs, each with one country having an unchanged capital and one with a changed capital. Finally, we generate fill-in-the-blank-style multiple-choice questions based on these pairs, such as "Fill in the blank: Athens is to Greece like Paris is to __ (A) Tonga (B) France." For this example, both the $r_{\texttt{cityOf}}$ and $r_{\texttt{capitalOf}}$ relations provide sufficient reasoning to answer as "France". While the corresponding synthetic explanation, $\varepsilon_{\texttt{capitalOf}}$, for the model with unaltered capitals would be "The capital of France is Paris, as the capital of Greece is Athens.", the one for the model with altered capitals, $\varepsilon_{\texttt{cityOf}}$, would be "Paris is a city in France, as Athens is a city in Greece."

## 4.3   OBJECT COUNTING TASKS

**Task**   Inspired by the `object_counting` task from BIG-bench (bench authors, 2023), we adapt an object counting task for evaluating diagnosticity. The task involves counting entities of a given type from a list of entities. By modifying model knowledge to swap objects across predefined categories, we ensure the number of entities of the target type remains the same while changing the reasoning behind the answer. For example, when asked how many of "countertop," "grape," and "kiwifruit" are fruits, the answer is 2, since "countertop" is a furniture item. If we edit the model to classify "countertop" as a fruit and "grape" as furniture, the answer remains 2 but due to different reasoning.

**Dataset**   We define five categories with five types each, as shown in Table 2 in Appendix A . For each type, we select 10 representative entities from WikiData. We then reserve 20% of the entities for reassignment to other types within the same category after model editing. We include two question types: yes/no questions, asking if all or any items in a list belong to a given type, and number questions, asking how many items belong to a specific type.

For both types, we randomly determine the number of items $k$ (between 3 and 6) and select a target type. For yes/no questions, we sample $k$ entities, ensuring that after model editing, the number of entities of the target type remains unchanged. For number questions, we reassign one entity from the target type and one from other types to ensure consistency.

We generate 1,000 samples in total, equally divided between the two question types. Further details about the dataset generation process are included in Appendix A.

---

[1] `https://www.kaggle.com/datasets/viswanathanc/world-cities-datasets/`

| | | Metric | Mistral-Instruct | LLaMa-2-7b-chat | LLaMa-2-7b | GPT-J 6B |
|---|---|---|---|---|---|---|
| **FactCheck** | **Posthoc** | CC-SHAP | **0.437** | **0.518** | **0.665** | **0.553** |
| | | Simulatability | 0.014 | 0.052 | 0.035 | 0.033 |
| | | Counterfact. Edits | 0.001 | 0.000 | 0.000 | 0.000 |
| | **CoT** | Early Answering | 0.030 | 0.033 | 0.045 | 0.056 |
| | | Filler Tokens | 0.019 | 0.029 | 0.025 | 0.022 |
| | | Adding Mistakes | 0.013 | 0.047 | 0.158 | 0.029 |
| | | Paraphrasing | 0.160 | 0.108 | 0.171 | 0.029 |
| | | CC-SHAP | **0.559** | **0.522** | **0.616** | **0.547** |
| **Analogy** | **Posthoc** | CC-SHAP | **0.850** | **0.583** | **0.657** | **0.355** |
| | | Simulatability | 0.006 | 0.001 | 0.000 | 0.000 |
| | | Counterfact. Edits | 0.001 | 0.000 | 0.000 | 0.000 |
| | **CoT** | Early Answering | 0.041 | 0.018 | 0.110 | 0.063 |
| | | Filler Tokens | 0.041 | 0.011 | 0.044 | 0.145 |
| | | Adding Mistakes | 0.118 | 0.023 | 0.190 | 0.198 |
| | | Paraphrasing | 0.123 | 0.121 | 0.165 | 0.235 |
| | | CC-SHAP | **0.859** | **0.663** | **0.672** | **0.411** |
| **Object Counting** | **Posthoc** | CC-SHAP | **0.522** | **0.460** | **0.510** | **0.500** |
| | | Simulatability | 0.031 | 0.028 | 0.037 | 0.034 |
| | | Counterfact. Edits | 0.000 | 0.000 | 0.000 | 0.000 |
| | **CoT** | Early Answering | 0.109 | 0.005 | 0.086 | 0.120 |
| | | Filler Tokens | 0.065 | 0.033 | 0.058 | 0.074 |
| | | Adding Mistakes | 0.124 | 0.129 | 0.109 | 0.164 |
| | | Paraphrasing | 0.191 | 0.173 | 0.154 | 0.190 |
| | | CC-SHAP | **0.504** | **0.467** | **0.494** | **0.509** |

Table 1: The diagnosticity scores of each model for each faithfulness metric across three tasks, along with the accuracy of each model on each task under standard and CoT prompting. Bold numbers indicate the highest scores for each model on each task across the two categories of faithfulness metrics: post-hoc and CoT. "Mistral-Instruct" refers to the `mistral-7b-instruct-v0.2` model.

## 5 EXPERIMENTS

We present four sets of experiments. First, we report the diagnosticity scores of post-hoc and CoT-based metrics across three tasks and four LLMs. Second, we conduct an analysis to assess the reliability of the model edits used for diagnosticity evaluation. Third, we perform an ablation study where we replace MEMIT with a simplified version of IKE, examining how the choice of model editing method affects our results. Finally, we conduct another ablation study in which we use model-generated explanations instead of synthetically generated ones.

### 5.1 DIAGNOSTICITY EVALUATION OF FAITHFULNESS METRICS

**Experimental Setup** We evaluate the seven metrics described in Section 2 across four different LLMs: `mistral-instruct-7b-v0.2` (Jiang et al., 2023), `llama-2-7b`, `llama-2-7b-chat` (Touvron et al., 2023), and `gpt-j-6B` (Wang & Komatsuzaki, 2021). For our main experiments, we employ MEMIT as the model editing method and use synthetic explanations to ensure their faithfulness with respect to the edited model.

Table 1 presents the diagnosticity scores for all faithfulness metrics across three tasks for the four models. The most notable finding is that CC-SHAP significantly outperforms other methods (McNemar's test, $p < .01$) in each task, for each model, across both post-hoc and CoT-based metrics. In the post-hoc category, Simulatability shows significantly higher diagnosticity than Counterfactual Edits across all models for the Object Counting and FactCheck tasks (McNemar's test, $p < .01$), and higher or comparable diagnosticity for the Analogy task. In the Analogy task, Paraphrasing and Adding Mistakes significantly outperform other CoT-based metrics (McNemar's test, $p < .01$),

following CC-SHAP, across all models with the exception of Adding Mistakes in `llama-2-7b-chat`. For the Object Counting task, Paraphrasing becomes the second-best CoT-based metric, significantly outperforming other metrics for all models (McNemar's test, $p < .01$) except `gpt-j-6b`.

Although CC-SHAP outperforms Paraphrasing by a wide margin, Paraphrasing consistently ranks as the second-highest diagnosticity metric in most cases, followed by Adding Mistakes, Early Answering, and Filler Tokens, respectively. However, there are some exceptions to this order. For instance, Early Answering ranks as the second-best metric in the FactCheck task for `gpt-j-6b`, while Adding Mistakes ranks second in the Analogy task for `llama-2-7b`. Although this ranking generally holds, the relative differences are not always statistically significant.

When examining cases where faithfulness metrics fail to correctly assign higher scores to faithful explanations, we find that binary metrics (all except CC-SHAP) often struggle to differentiate between the faithfulness levels of explanations, frequently assigning the same score to both. Across all three tasks, most binary metrics fail in this manner at least 90% of the time. However, some metrics more frequently assign lower scores to faithful explanations than to unfaithful ones. For example, *Paraphrasing* assigns lower scores to faithful explanations at least 15% of the time across all tasks, while *Adding Mistakes* and *Early Answering* do so at least 15% of the time for the Object Counting task. A closer look at *Paraphrasing* examples reveals that the paraphrasing process can lead to significant hallucinations, sometimes even causing paraphrases of contradictory explanation pairs to state the same facts.

**These findings highlight the importance of carefully selecting a helper model when using faithfulness metrics based on corrupting CoT.** Following Parcalabescu & Frank (2023), we use `llama-2-13b-chat` as our helper model. Similarly, Lanham et al. (2023) use the same model as their predictor and explainer: a 175B-parameter decoder-only transformer LLM (Vaswani et al., 2017; Radford & Narasimhan, 2018; Radford et al., 2019; Brown

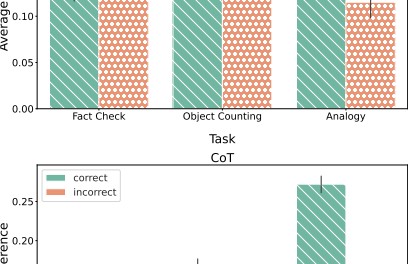

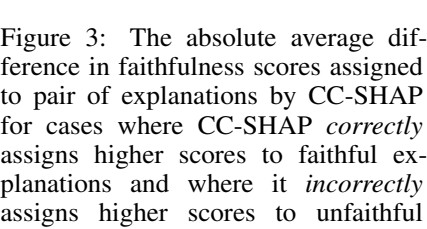

Figure 3: The absolute average difference in faithfulness scores assigned to pair of explanations by CC-SHAP for cases where CC-SHAP *correctly* assigns higher scores to faithful explanations and where it *incorrectly* assigns higher scores to unfaithful ones, across all tasks, for both post-hoc and CoT-based CC-SHAP, using `mistral-7b-instruct-v0.2`

et al., 2020). While these issues may be less apparent with larger models, practitioners should be cautious when using a helper model of similar size to the model being tested, particularly for smaller models.

Since CC-SHAP is a smoother metric, we find no instances where it fails by assigning the same score to both explanations. To gain deeper insight, we examine the average absolute differences in faithfulness scores between each pair of explanations. Figure 3 presents these differences for cases where CC-SHAP correctly assigns higher scores to faithful explanations and where it incorrectly assigns higher scores to unfaithful ones, across all tasks, for both post-hoc and CoT-based CC-SHAP, using `mistral-7b-instruct-v0.2`. The results indicate that the average absolute differences in faithfulness scores are generally similar for both correct and incorrect cases. However, in the Analogy task, CC-SHAP better distinguishes between faithful and unfaithful explanations when it performs correctly compared to when it fails. While this observation aligns with the task in which CC-SHAP achieves its highest diagnosticity scores, no significant correlation is found between diagnosticity and the average absolute difference in faithfulness scores.

## 5.2 RELIABILITY OF EDITS

CAUSAL DIAGNOSTICITY relies on the assumption that, in each given explanation pair, one explanation is faithful to the model being evaluated while the other is unfaithful. To ensure this condition is

met, we modify the models and use synthetically generated explanation pairs. While these synthetic explanations logically guarantee faithfulness or unfaithfulness with respect to the edited model, their practical accuracy depends on the success of the editing method. One way to assess whether the synthetic explanations align with faithfulness expectations is by comparing the perplexities of the explanation pairs. Since the only difference between the explanations is related to the aspect modified by the model edit, the intuition is that the explanation deemed faithful should have a lower perplexity than the one deemed unfaithful.

Figure 4 shows the frequency with which explanations deemed as faithful have lower perplexity than those deemed as unfaithful, for each task and each model. While the explanations deemed faithful generally have lower perplexities than their unfaithful counterparts across all tasks, the edits performed for the Fact Check task are particularly successful, with scores nearing 1.0. In contrast, the edits for the Analogy and Object Counting tasks perform relatively worse. The scores for these two tasks are similar across all models, except for `gpt-j-6b`, where the edits for the Analogy task perform notably worse.

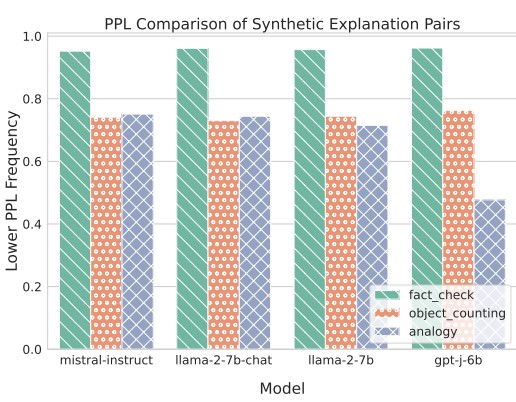

Figure 4: Frequency of which explanations deemed as faithful have lower perplexity than those deemed as unfaithful, for each task and each model. Higher frequency indicates the higher success in applied edits.

### 5.3 EFFECT OF KNOWLEDGE EDITING METHOD

We investigate the effect of different model editing methods on our results by conducting an ablation study where we replace MEMIT with an alternative approach. Instead of selecting another locate-then-edit method, we use a simplified version of IKE Zheng et al. (2023), a memory-based knowledge editing technique. Details of this method are provided in Appendix B.

Figure 5 compares MEMIT and IKE across all faithfulness metrics, with diagnosticity scores averaged over three tasks. While the diagnosticity scores from models edited with MEMIT are higher than those obtained with IKE, the relative relationships between different metrics remain consistent. This suggests that the choice of model editing method has no significant impact on our conclusions.

### 5.4 EFFECT OF MODEL GENERATED EXPLANATIONS

While our main results are derived from using synthetically generated explanations to form faithful and unfaithful explanation pairs that accurately reflect the applied edits and the differences between the models, we also perform an ablation study using model-generated explanations.

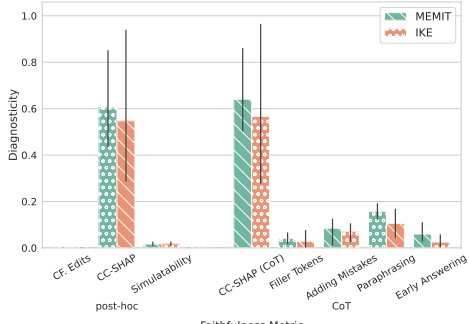

Figure 5: Diagnosticity scores for each metric on `mistral-7B-instruct-v0.2` using two model editing methods: MEMIT and IKE. Although the scores are higher when MEMIT is used, the ranking of the metrics remains consistent across both editing methods.

**Experimental Setup**   We evaluate all faithfulness metrics using `mistral-7B-instruct-v0.2`. For model-generated explanations, the length is limited to 100 tokens.

Figure 6 compares model-generated and synthetic explanations across all faithfulness metrics, with diagnosticity scores averaged over three tasks. Although CC-SHAP consistently outperforms both other post-hoc and CoT-based metrics for both explanation types, there is no consistent difference in the diagnosticity scores between the two explanation types across all metrics. Furthermore, the

comparative ranking of faithfulness metrics is inconsistent when replacing synthetic explanations with model-generated ones. Upon examining the model-generated explanations, we observe several issues. At times, explanations pairs contain hallucinations, making them unfaithful to their own models and violating the main condition of our framework. Occasionally, explanations are truncated due to the token limit. In some cases, an explanation may begin appropriately but revert to pre-edit knowledge. Particularly in the Analogy and Object Counting tasks, models often fail to articulate the applied edits. While these issues could be attributed to the limited generalizability of model editing methods, larger models or memory-based editing approaches may help address these challenges (Yao et al., 2023). Nevertheless, synthetically generated explanations stand out as a viable option, especially when considering the computational costs associated with these alternatives.

## 6 CONCLUSION

In this paper, we introduce a new framework, CAUSAL DIAGNOSTICITY, to evaluate faithfulness metrics for natural language explanations by extending the notion of diagnosticity. We introduce three new tasks—fact-checking, analogy, and object counting—while utilizing model editing to generate pairs of faithful and unfaithful explanations to measure diagnosticity. We benchmark popular post-hoc and CoT-based faithfulness metrics across these tasks. The results show that most metrics fail to achieve satisfactory diagnosticity scores, with CC-SHAP being a notable exception. Unlike other methods, CC-SHAP leverages more information by considering token-wise interactions between the explanation and the input, which likely allows it to capture the inner workings of the model better than methods that simply observe changes in output after perturbing the input or explanations. Despite CC-SHAP's higher scores, the results

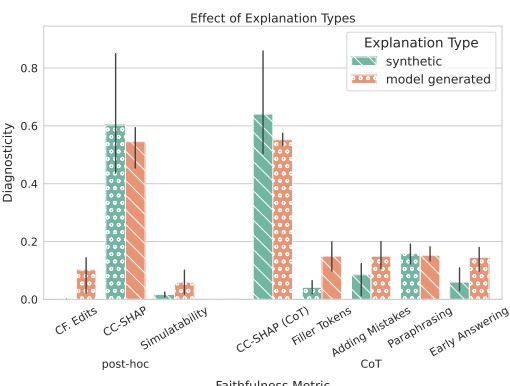

Figure 6: Diagnosticity scores for each metric on `mistral-7B-instruct-v0.2` using model generated and synthetically generated explanations.

also highlight areas for improvement, particularly in terms of the computational cost and slowness of CC-SHAP. Based on these findings, developing metrics that focus more on the model's internal mechanisms and complex interactions among explanations, inputs, and outputs could be a promising direction.

We view this study as the first step in the quest for more faithful LLM explanations by providing a test bed for faithfulness metrics. As our study reveals the inadequacy of existing metrics and underscores the need for better alternatives, a natural direction for future research is the development of improved faithfulness metrics, which should then be followed by the creation of more faithful explanation methods.

## 7 LIMITATIONS

This study is limited to 7B-parameter models due to the availability of models with published hyperparameters for MEMIT editing and computational constraints. Additionally, CAUSAL DIAGNOSTICITY is heavily relies on the effectiveness of the model editing method. While we conduct an ablation study using the IKE baseline, the utility of model-generated explanations remains largely unexplored. This is because approaches to address issues in model-generated explanations, such as employing memory-based knowledge editing methods or using larger models, come with high computational costs. In particular, memory-based methods lead to lengthy experiments with CC-SHAP due to the increased context length.

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

## A  DATASET

Figure 7 illustrates the prompt used to convert statements from COUNTERFACT into yes/no questions for the Fact Check task, utilizing `Mistral-7B-Instruct-v0.2`. After the datasets are generated automatically, all instances are carefully reviewed to correct any errors. Table 2 presents the categories and types used in the Object Counting task.

## B  MODEL EDITING

### B.1  TASK-BASED EDITING TEMPLATES

Table 3 shows the templates we use for editing models in each task. For the FactCheck task, there is a variety of prompts where the action or situation of the subject differs, but the target is always located

| Category | Types |
|---|---|
| object | animal, musical instrument, fruit, vegetable, furniture |
| occupation | scientist, politician, soccer player, actor, singer |
| company | media company, energy company, software company, automotive company, consulting company |
| touristic attraction | France, Spain, Russia, Turkey, Italy |
| abstract | religion, political ideology, language, branch of science, emotion |

Table 2: Categories and corresponding types used in Object Counting task

```
Please create a yes-no question from the given sentence. Here are some examples:
Sentence: Joe Biden is the president of the United States. Question: Is Joe Biden the
    ↪ president of the United States?
Sentence: They play rock. Question: Do they play rock?
Sentence: Quesadilla from Mexico. Question: Is quesadilla from Mexico?
Do not mention your assumptions or assesment towards correctness of question. Do not output
    ↪ anything else! Stick with the format.
Sentence: {SENTENCE} Question:
```

Figure 7: The prompt used for converting statements to questions.

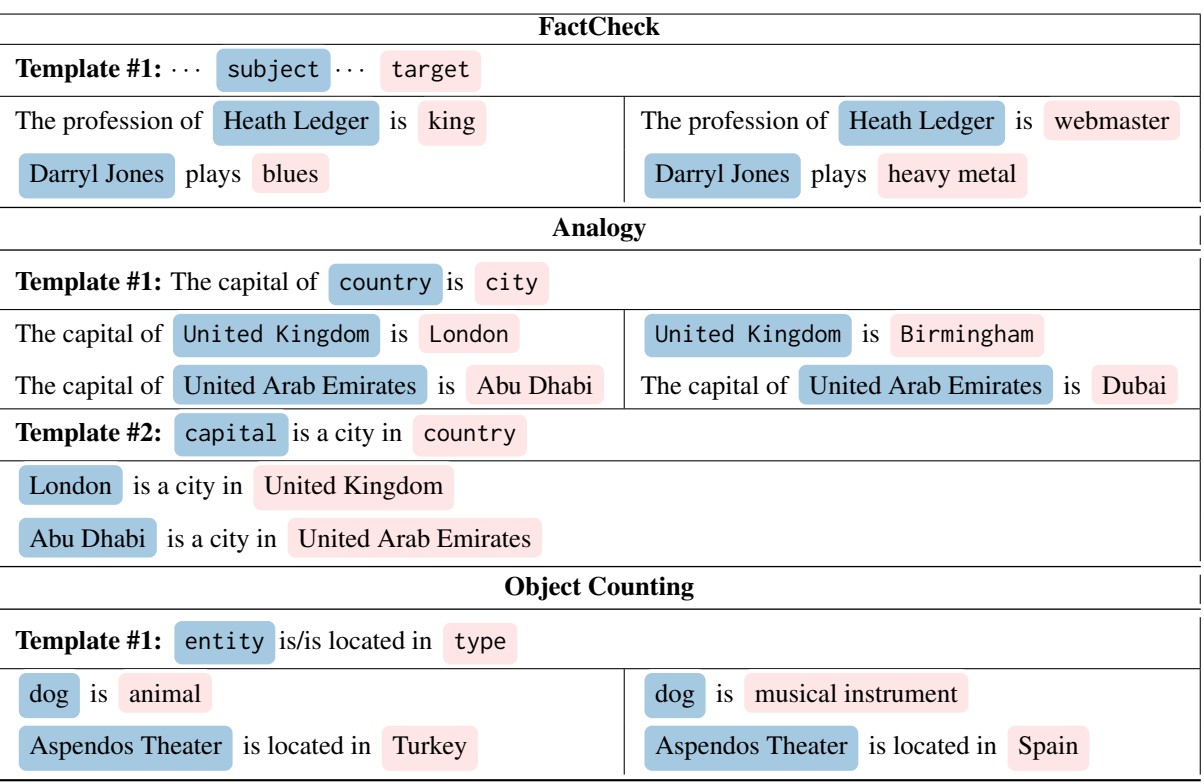

Table 3: Templates used for editing models. Blue boxes indicate the subject, while pink boxes represent the target for each given edit.

at the end of the prompt. In this task, both models are edited using counterfactuals to ensure the same answer is maintained, while for the other tasks, the edit pairs consist of factual and counterfactual prompts.

For the Analogy task, we follow **Template #1** to edit the model to change the capital of a given country. Even for the model where the capitals remain unchanged, we apply this edit in case the

model lacks knowledge of some countries. For both models, we reinforce the $r_{\texttt{cityOf}}$ relation by applying **Template #2**.

For the Object Counting task, we use the corresponding template in Table 3 to edit the model by altering the types of entities. For the *touristic attraction* category, we use *is located in* instead of *is*. Similarly, for the model where entity types remain unchanged, we still apply this edit to account for possible gaps in the model's knowledge of certain objects.

## B.2 IN-CONTEXT KNOWLEDGE EDITING (IKE) BASELINE

Zheng et al. (2023) leverage In-Context Learning for knowledge editing in LLMs without requiring parameter updates. They define three types of in-context demonstrations to enhance generalization (the ability to update knowledge expressed in different textual forms) and specificity (the ability to avoid altering unrelated knowledge when making edits). These demonstrations are: (1) *copy* for injecting new facts, (2) *update* for improving generalization, and (3) *retain* for preventing changes to unrelated knowledge. However, for our IKE experiments, we adopt their simpler PROMPT baseline, where new facts are directly added to the context. We use the same templates shown in Table 3, but prepend each relevant edit just before the query. When measuring faithfulness scores, we exclude the prefix containing these edits from any operations and keep it fixed.

## C ADDITIONAL RESULTS

### C.1 IKE RESULTS

| | Metric | FactCheck | Analogy | Object Counting |
|---|---|---|---|---|
| **Posthoc** | CC-SHAP | **0.418** | **0.938** | **0.287** |
| | Simulatability | 0.014 | 0.008 | 0.032 |
| | Counterfact. Edits | 0.000 | 0.000 | 0.000 |
| **CoT** | Early Answering | 0.016 | 0.003 | 0.057 |
| | Filler Tokens | 0.011 | 0.001 | 0.075 |
| | Adding Mistakes | 0.027 | 0.085 | 0.104 |
| | Paraphrasing | 0.105 | 0.046 | 0.167 |
| | CC-SHAP | **0.460** | **0.963** | **0.279** |

Table 4: The diagnosticity scores of `mistral-7b-instruct-v0.2` for each faithfulness metric across three tasks, along with the accuracy of each model on each task under standard and CoT prompting when IKE baseline is used as model editing method. Bold numbers indicate the highest scores for each model on each task across post-hoc and CoT-based faithfulness metrics.

