# OpenReview forum: "A Causal Lens for Evaluating Faithfulness Metrics"
_ICLR.cc/2025/Conference — Submitted to ICLR 2025_

### Official Review · Reviewer_poTs · 2024-10-31

**Soundness:** 2
**Presentation:** 4
**Contribution:** 2
**Rating:** 3
**Confidence:** 4

**Summary:**

This paper presents a new framework to assess existing faithfulness metrics for natural language explanations (NLEs) from Large Language Models (LLMs). It adapts a previous test for faithfulness metrics, diagnosticity, which uses random feature attributions as unfaithful explanations. The new framework, causal diagnosticity, edits the LLM's knowledge in order to generate explanations that can be determined as unfaithful. Several existing metrics are evaluated on modified datasets across a range of tasks.

**Strengths:**

- Tailors existing datasets to the new tasks.
- Comprehensive evaluation across a range of tasks, with many existing metrics.
- The paper is well written. The main idea of using two models for causal diagnosticity is emphasized throughout. Separate applications are detailed well.
- Decent evidence is provided to demonstrate that many existing faithfulness metrics are staggeringly weak, as they fail to pass tests on basic tasks.

**Weaknesses:**

- The paper points out that many existing faithfulness metrics from [1] are flawed, which is useful extra evidence, though it is fundamentally unclear to me how this is going to lead to substantial later improvements.
- While the datasets used are systematic enough to gain insights, a big fear is that they are overly simplistic. The examples given are for very short chain-of-thought responses to basic questions. Moreover, most of the faithfulness metrics in [1] are quite premature and have been criticized in the literature [2, 3]. It makes sense to me therefore that CC-SHAP might perform well, but the performance here does not really shed insight into the performance of the metric on more subtle scenarios (nor is it clear how well the evaluation framework is able to handle real-world tasks such as medical question answering).
- These are my fundamental issues with the current setup, alongside relatively small/similar models being assessed. The faithfulness problem is due mostly to the fact that we do not have good ground truth of what faithful explanations are internally. This paper offers ground truths but in very simple settings.

[1] Measuring Faithfulness in Chain-of-Thought Reasoning, Lanham et. al., 2023

[2] Chain-of-Thought Unfaithfulness as Disguised Accuracy, Bentham et. al., 2024

[3] On Measuring Faithfulness or Self-consistency of Natural Language Explanations, Parcalabescu and Frank, 2023

**Questions:**

- Regarding model editing, is changing a single fact (i.e. “is paris the capital of france”) sufficient to override all pretrained knowledge in the LLM? I ask because this is a key part of the evaluation framework.
- How might this framework, especially model editing aspects, be adapted to more complex/realistic scenarios?
- Minor: I may be misunderstanding something here, but do you need to modify two models for causal diagnosticity? Is it not sufficient to change the knowledge in one LLM, and compare the modified LLM to the original?

---

### Official Review · Reviewer_9PLF · 2024-11-03

**Soundness:** 2
**Presentation:** 2
**Contribution:** 2
**Rating:** 5
**Confidence:** 4

**Summary:**

The authors propose **Causal Diagnosticity**, a metric designed to evaluate the diagnosticity of existing faithfulness metrics for natural language explanations, where diagnosticity indicates how often a faithfulness metric favors faithful explanations over unfaithful ones.

Through evaluations of several post-hoc and Chain of Thought (CoT) faithfulness metrics across four tasks and four large language models (LLMs), the authors conclude that CC-SHAP outperforms the other metrics.

**Strengths:**

- Natural language explanations are becoming increasingly relevant due to advancements in LLMs, making them more accessible for end users. This work aids users in selecting a metric to evaluate the faithfulness of natural language explanations.
- The idea of extending diagnosticity from feature attribution to natural language explanations is straightforward and relevant.
- The authors conducted empirical experiments and discussed the effects of model knowledge editing methods, explanation types, and the reliability of these edits. The results demonstrate that CC-SHAP outperforms other metrics.

**Weaknesses:**

1. In Equation (5), the assumption is made that $\overline\epsilon_i$ is faithful to $\overline\theta$, while $\widetilde\epsilon_i$ is not. However, this is not guaranteed in the experiments. Though the authors discuss this in Section 5.2, it's unclear why perplexity can indicate faithfulness.


2. The model editing discussed in Section 2.2 involves modifying the internal weights of LLMs, which restricts the experiments to open-source LLMs, excluding closed-source models like GPT-4.

3. The model $\widetilde\theta$ is exclusively used to generate unfaithful explanations and is not incorporated in Equation (5). Since directly modifying $\overline\epsilon$ could also yield an unfaithful explanation related to $\overline\theta$, the necessity of using $\widetilde\theta$ is unclear.

4. The placement of figures throughout the paper is disorganized, which hurts readability.

**Questions:**

1. The following questions are about perplexity described in Section 5.2:
    - What is perplexity? Could the authors provide a definition and explain how it is calculated?
    - Why can low perplexity indicate a faithful explanation?
    - If low perplexity does indeed correlate with faithful explanations, then perplexity itself could serve as a metric for evaluating faithfulness. Why is it not evaluated with **Causal Diagnosticity**? Without assessing the **Causal Diagnosticity** of perplexity, how can it be sufficient to determine whether an explanation is faithful?

2. Are the synthetically generated explanations produced by the LLMs themselves? In line 203, the authors mention that "$\overline\theta$ generates the explanation $\overline\epsilon$ and $\widetilde\theta$ generates the explanation $\widetilde\epsilon," which is confusing.

3. Is the use of $\widetilde\theta$ necessary?

---

### Official Review · Reviewer_Gz5g · 2024-11-03

**Soundness:** 1
**Presentation:** 2
**Contribution:** 2
**Rating:** 3
**Confidence:** 4

**Summary:**

This work proposes Causal Diagnoticity, a benchmark to evaluate metrics that evaluate natural language explanations generated by LLMs. In particular, this benchmark creates carefully edited models for three tasks that lead to unusual reasoning in three tasks: fact-checking, analogy and object counting. For each model response, two explanations are synthetically generated, with one containing the correct reasoning and the other containing the incorrect reasoning. Then different faithfulness metrics are computed to evaluate this pair of explanations, and the score difference between the correct and incorrect explanations is taken as the quality (i.e., diagnosticity) of the metric. Experimental results suggest that CC-Shap is the best performing, but still leaves much room for improvement.

**Strengths:**

The framework on evaluating faithfulness metrics for natural language explanations is quite novel.

The use of model editing to create the three synthetic tasks is also very novel.

Extensive evaluations of several different faithfulness metrics are used.

**Weaknesses:**

My biggest concern is with the generation of synthetic explanations, and the assumption that one is correct and the other is incorrect. In particular, while the model is edited on the particular fact, it is unclear that the particular editing causes the model to use the "intended" reasoning path, or the model is actually using some very different reasoning paths. For example, in the Rihanna example, it could be that the model editing removes "Rihanna" entity from the "singer set", and hence results in the model predicting no. In this case, the "correct" explanation should be something like "No, because I do not find Rihanna in the singer set" (though this language may be highly unlikely to be generated by an LLM). How to carefully eliminate such possibilities is, in my opinion, the most crucial issue in establishing the soundness of the framework.

In addition, for a slightly related question, can we really assess the faithfulness of natural language explanations that are not generated by the models themselves (i.e., having low or almost-zero probability of being decoded), or in other words, is such evaluation fundamentally meaningful? In "traditional" interpretability, all the explanations, such as feature attribution, concept and counterfactual, are generated using well-defined algorithms, and hence the "meaningfulness" of the explanation is of little doubt. However, with synthetic explanations, I am less sure, so I would hope that the authors could convince me on this aspect.

Of a minor note, there are also alternative ways to generate and evaluate LLM-generated explanations, such as [1] and [2], that could be discussed in related work. Furthermore, there is even work that outright assert that LLMs cannot explain via natural language explantions [3].

[1] https://arxiv.org/abs/2310.11207

[2] https://aclanthology.org/2024.findings-acl.19.pdf

[3] https://arxiv.org/pdf/2405.04382

**Questions:**

See weaknesses.

---

### Official Review · Reviewer_bgKV · 2024-11-03

**Soundness:** 3
**Presentation:** 2
**Contribution:** 2
**Rating:** 5
**Confidence:** 3

**Summary:**

The paper presents an empirical study to analyse the **reliability of faithfulness metrics for explanations of LLMs**. The authors propose "Causal Diagnosticity" as a metric to evaluate seven faithfulness metrics (for both post-hoc and Chain of Thought explanations) on three NLP tasks across four different LLMs. The results suggest that CC-SHAP is the most reliable metric among all the evaluated faithfulness metrics.

**Strengths:**

1. The paper is well-written - the motivation of the work is clearly presented, related works are well discussed, proposed approach and experiments are clearly described, and results are well discussed.

2. The topic the paper focuses on is extremly important. Given the widespread usage of LLMs, it is very important to develop faithful methods to explain their predictions, but it is equally important to benchmark them.

3. The experiments are diverse and include ablation studies to understand if and how the conclusion generalises.

**Weaknesses:**

1. The paper seems very applied to me with limited novelty. The authors expand an existing metric (called diagnosticity) to natural language explanations by arguing that random text cannot work as meaningful explanation (line 188..). However, this argument needs more backing/examples as random text can be considered as unfaithful explanation as done previously by Chan et al. 2022b.

2. Secondly, the authors introduce model editing as a way to generate pair of explanations (faithful and unfaithful), but this might be limiting the analysis as a given model editing method may not work perfectly (because the knowledge may be incorrectly learned), hence, model prediction may change, but the reason is incorrect. The authors argue to use "synthetic explanations" to handle this, but it is not clear if insights on synthetic explanations are generalisable to the real world. For e.g., are the syntetic explanations guranteed to not hallucinate?

**Questions:**

1. Can the author say more on how do they generate synthetic explanations?
2. Can the authors justify the use of word "causal" in the metric/framework they introduce?
3. Fig. 6, seems to have a bug in the legend?

---

### Official Review · Reviewer_PpGV · 2024-11-12

**Soundness:** 3
**Presentation:** 3
**Contribution:** 3
**Rating:** 6
**Confidence:** 3

**Summary:**

The paper proposes a new framework to assess the faithfulness of natural language explanations generated by large language models (LLMs). To evaluate existing metrics, the framework employs model editing to create pairs of faithful and unfaithful explanations and tests various metrics using a benchmark of three tasks: fact-checking, analogy, and object counting. The study finds that while the CC-SHAP metric consistently outperforms others, many metrics fail to reliably capture faithfulness.

**Strengths:**

1. Originality: The paper introduces a novel approach that uses causal model editing to generate faithful-unfaithful explanation pairs, offering a rigorous basis for assessing faithfulness in natural language explanations. This approach combines causality with faithfulness evaluation and tries to get to the model’s true reasoning processes.

2. Quality: The paper is rigorous, with comprehensive experiments across three tasks and multiple language models. The inclusion of alternative model editing techniques and ablation studies further strengthens the evaluation framework.

3. Clarity: The paper is well-organized, with clear, detailed explanations. Tables and figures present complex concepts in easy to understand manner.

4. Significance: This work is significant in that its findings highlight specific improvements needed in LLM explanation fidelity.

In summary, this paper is a good contribution to the field of LLM interpretability.

**Weaknesses:**

1. The use of synthetic explanations may be limiting, as these pairs might not fully represent actual model-generated explanations. It would be helpful if the authors provided an analysis of how well synthetic explanations align with actual ones.
2. The focus on three specific tasks (fact-checking, analogy, object counting) may not generalize well to more complex contexts. Adding diverse tasks or discussing broader applicability would be helpful. Have the authors considered experimenting with other complex contexts?
3. Relying on diagnosticity as a faithfulness measure is overlooking other aspects of reasoning, like consistency and coherence. Including complementary metrics or discussing proposed framework’s limitations would be helpful.

**Questions:**

See above in weaknesses.

---

### Meta-Review · Area_Chair_rXhX · 2024-12-20

**Metareview:**

This paper develops new way to understand faithfulness in natural language explanations through causality. Their framework causal diagnosticity provides a main for contrasting different failthfulness evaluations. They also introduce a benchmark and evaluate existing faithfulness measures.  The reviewers wore mostly negative with two rejects. The concerns ranged from criticism of faithfulness metrics existing in previous works, the complexity of the benchmark, and the readability of the paper.

**Additional Comments On Reviewer Discussion:**

There was no author response

---

### Decision · Program_Chairs · 2025-01-22

Reject